# Fooling Neural Network Interpretations via Adversarial Model Manipulation

**Juyeon Heo**[1],[*] **Sunghwan Joo**[1],[*] **and Taesup Moon**[1,2]
[1]Department of Electrical and Computer Engineering, [2]Department of Artificial Intelligence
Sungkyunkwan University, Suwon, Korea, 16419
heojuyeon12@gmail.com, {shjoo840, tsmoon}@skku.edu

## Abstract

We ask whether the neural network interpretation methods can be fooled via adversarial model manipulation, which is defined as a model fine-tuning step that aims to radically alter the explanations without hurting the accuracy of the original models, e.g., VGG19, ResNet50, and DenseNet121. By incorporating the interpretation results directly in the penalty term of the objective function for fine-tuning, we show that the state-of-the-art saliency map based interpreters, e.g., LRP, Grad-CAM, and SimpleGrad, can be easily fooled with our model manipulation. We propose two types of fooling, Passive and Active, and demonstrate such foolings generalize well to the entire validation set as well as transfer to other interpretation methods. Our results are validated by both visually showing the fooled explanations and reporting quantitative metrics that measure the deviations from the original explanations. We claim that the stability of neural network interpretation method with respect to our adversarial model manipulation is an important criterion to check for developing robust and reliable neural network interpretation method. The source code is available at https://github.com/rmrisforbidden/FoolingNeuralNetwork-Interpretations.

## 1   Introduction

As deep neural networks have made a huge impact on real-world applications with predictive tasks, much emphasis has been set upon the interpretation methods that can explain the ground of the predictions of the complex neural network models. Furthermore, accurate explanations can further improve the model by helping researchers to debug the model or revealing the existence of unintended bias or effects in the model [1, 2]. To that regard, research on the interpretability framework has become very active recently, for example, [3, 4, 5, 6, 7, 8, 9], to name a few. Paralleling above flourishing results, research on sanity checking and identifying the potential problems of the proposed interpretation methods has also been actively pursued recently. For example, some recent research [10, 11, 12, 13, 14] showed that many popular interpretation methods are not stable with respect to the perturbation or the adversarial attacks on the *input* data.

In this paper, we also discover the instability of the neural network interpretation methods, but with a fresh perspective. Namely, we ask whether the interpretation methods are stable with respect to the *adversarial model manipulation*, which we define as a model fine-tuning step that aims to dramatically alter the interpretation results without significantly hurting the accuracy of the original model. In results, we show that the state-of-the-art interpretation methods are vulnerable to those manipulations. Note this notion of stability is clearly different from that considered in the above mentioned works, which deal with the stability with respect to the perturbation or attack on the input to the model. To the best of our knowledge, research on this type of stability has not been explored before. We believe that such stability would become an increasingly important criterion to check,

---

[*]Equal contribution.

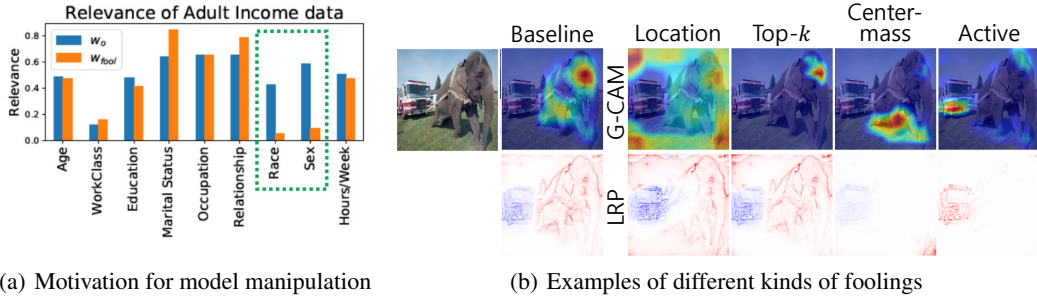

(a) Motivation for model manipulation                    (b) Examples of different kinds of foolings

Figure 1: (a) The result of our fooling on the 'Adult income' classification data [15]. We trained a classifier with 8 convolution layers, $w_o$, and the LRP result (blue) shows it assigns high importance on *sensitive* features like 'Race' and 'Sex'. Now, we can manipulate the model with Location fooling (See Section 3) that zero-masks the two features and obtain $w_{\text{fool}}^*$ that essentially has the same accuracy as $w_o$ but with a new interpretation that disguises the bias (orange). (b) The interpretation results for the image [16] on the left with prediction "Indian Elephant". **The first column** is for the original pre-trained VGG19 model, **the second to fourth column** are for the six manipulated models with *Passive foolings* (highlighting uninformative pixels of the image), and **the fifth column** is for the two manipulated models with *Active fooling* (highlighting a completely different object, the firetruck). **Each row** corresponds to the interpretation method used for fooling. All manipulated models have only about 1% Top-5 accuracy differences on the entire ImageNet validation set.

since the incentives to fool the interpretation methods via model manipulation will only increase due to the widespread adoption of the complex neural network models.

For a more concrete motivation on this topic, consider the following example. Suppose a neural network model is to be deployed in an income prediction system. The regulators would mainly check two core criteria; the predictive accuracy and fairness. While the first can be easily verified with a holdout validation set, the second is more tricky since one needs to check whether the model contains any unfair bias, e.g., using *race* as an important factor for the prediction. The interpretation method would obviously become an important tool for checking this second criterion. However, suppose a lazy developer finds out that his model contains some bias, and, rather than actually fixing the model to remove the bias, he decides to manipulate the model such that the interpretation can be fooled and hide the bias, without any significant change in accuracy. (See Figure 1(a) for more details.) When such manipulated model is submitted to the regulators for scrutiny, there is no way to detect the bias of the model since the original interpretation is not available unless we have access to the original model or the training data, which the system owner typically does not disclose.

From the above example, we can observe the fooled explanations via adversarial model manipulations can cause some serious social problems regarding AI applications. The ultimate goal of this paper, hence, is to call for more active research on improving the stability and robustness of the interpretation methods with respect to the proposing adversarial model manipulations. The following summarizes the main contributions of this paper:

- We first considered the notion of stability of neural network interpretation methods with respect to the proposing *adversarial model manipulation*.
- We demonstrate that the representative saliency map based interpreters, i.e., LRP [6], Grad-CAM [7], and SimpleGradient [17], are vulnerable to our model manipulation, where the accuracy drops are around 2% and 1% for Top-1 and Top-5 accuracy on the ImageNet validation set, respectively. Figure 1(b) shows a concrete example of our fooling.
- We show the fooled explanation *generalizes* to the entire validation set, indicating that the interpretations are truly fooled, not just for some specific inputs, in contrast to [11, 13, 14].
- We demonstrate that the *transferability* exists in our fooling, e.g., if we manipulate the model to fool LRP, then the interpretations of Grad-CAM and Simple Gradient also get fooled, etc.

## 2   Related Work

**Interpretation methods**   Various interpretability frameworks have been proposed, and they can be broadly categorized into two groups: black-box methods [18, 19, 5, 4, 20] and gradient/saliency map

based methods [6, 7, 21, 22, 23]. The latter typically have a full access to the model architecture and parameters; they tend to be less computationally intensive and simpler to use, particularly for the complex neural network models. In this paper, we focus on the gradient/saliency map based methods and check whether three state-of-the-art methods can be fooled with adversarial model manipulation.

**Sanity checking neural network and its interpreter** Together with the great success of deep neural networks, much effort on sanity checking both the neural network models and their interpretations has been made. They mainly examine the *stability* [24] of the model prediction or the interpretation for the prediction by either perturbing the input data or model, inspired by adversarial attacks [25, 26, 27]. For example, [10] showed that several interpretation results are significantly impacted by a simple constant shift in the input data. [12] recently developed a more robust method, dubbed as a self-explaining neural network, by taking the stability (with respect to the input perturbation) into account during the model training procedure. [11, 14] has adopted the framework of adversarial attack for fooling the interpretation method with a slight *input* perturbation. [13] tries to find perturbed data with similar interpretations of benign data to make it hard to be detected with interpretations. A different angle of checking the stability of the interpretation methods has been also given by [28], which developed simple tests for checking the stability (or variability) of the interpretation methods with respect to model parameter or training label randomization. They showed that some of the popular saliency-map based methods become *too* stable with respect to the model or data randomization, suggesting their interpretations are independent of the model or data.

**Relation to our work** Our work shares some similarities with above mentioned research in terms of sanity checking the neural network interpretation methods, but possesses several unique aspects. Firstly, unlike [11, 13, 14], which attack each given input image, we *change the model* parameters via fine-tuning a pre-trained model, and do *not* perturb the input data. Due to this difference, our adversarial model manipulation makes the fooling of the interpretations generalize to the entire validation data. Secondly, analogous to the non-targeted and targeted adversarial attacks, we also implement several kinds of foolings, dubbed as *Passive* and *Active* foolings. Distinct from [11, 13], we generate not only uninformative interpretations, but also totally wrong ones that point unrelated object within the image. Thirdly, as [12], we also take the explanation into account for model training, but while they define a special structure of neural networks, we do usual back-propagation to update the parameters of the given pre-trained model. Finally, we note [28] also measures the stability of interpretation methods, but, the difference is that our adversarial perturbation maintains the accuracy of the model while [28] only focuses on the variability of the explanations. We find that an interpretation method that passed the sanity checks in [28], e.g., Grad-CAM, also can be fooled under our setting, which calls for more solid standard for checking the reliability of interpreters.

## 3 Adversarial Model Manipulation

### 3.1 Preliminaries and notations

We briefly review the saliency map based interpretation methods we consider. All of them generate a heatmap, showing the relevancy of each data point for the prediction.

**Layer-wise Relevance Propagation (LRP) [6]** is a principled method that applies relevance propagation, which operates similarly as the back-propagation, and generates a heatmap that shows the *relevance value* of each pixel. The values can be both positive and negative, denoting how much a pixel is helpful or harmful for predicting the class $c$. In the subsequent works, LRP-Composite [29], which applies the basic LRP-$\epsilon$ for the fully-connected layer and LRP-$\alpha\beta$ for the convolutional layer, has been proposed. We applied LRP-Composite in all of our experiments.

**Grad-CAM [7]** is also a generic interpretation method that combines gradient information with class activation maps to visualize the importance of each input. It is mainly used for CNN-based models for vision applications. Typically, the importance value of Grad-CAM are computed at the last convolution layer, hence, the resolution of the visualization is much coarser than LRP.

**SimpleGrad (SimpleG) [17]** visualizes the gradients of prediction score with respect to the input as a heatmap. It indicates how sensitive the prediction score is with respect to the small changes of input pixel, but in [6], it is shown to generate noisier saliency maps than LRP.

**Notations** We denote $\mathcal{D} = \{(\mathbf{x}_i, y_i)\}_{i=1}^n$ as a supervised training set, in which $\mathbf{x}_i \in \mathbb{R}^d$ is the input data and $y_i \in \{1, \ldots, K\}$ is the target classification label. Also, denote $\boldsymbol{w}$ as the parameters for a

neural network. A heatmap generated by a interpretation method $\mathcal{I}$ for $\boldsymbol{w}$ and class $c$ is denoted by

$$\mathbf{h}_c^{\mathcal{I}}(\boldsymbol{w}) = \mathcal{I}(\mathbf{x}, c; \boldsymbol{w}), \tag{1}$$

in which $\mathbf{h}_c^{\mathcal{I}}(\boldsymbol{w}) \in \mathbb{R}^{d_{\mathcal{I}}}$. If $d_{\mathcal{I}} = d$, the $j$-th value of the heatmap, $h_{c,j}^{\mathcal{I}}(\boldsymbol{w})$, represents the importance score of the $j$-th input $x_j$ for the final prediction score for class $c$.

### 3.2 Objective function and penalty terms

Our proposed adversarial model manipulation is realized by fine-tuning a pre-trained model with the objective function that combines the ordinary classification loss with a penalty term that involves the interpretation results. To that end, our overall objective function for a neural network $\boldsymbol{w}$ to minimize for training data $\mathcal{D}$ with the interpretation method $\mathcal{I}$ is defined to be

$$\mathcal{L}(\mathcal{D}, \mathcal{D}_{fool}, \mathcal{I}; \boldsymbol{w}, \boldsymbol{w}_0) = \mathcal{L}_C(\mathcal{D}; \boldsymbol{w}) + \lambda \mathcal{L}_{\mathcal{F}}^{\mathcal{I}}(\mathcal{D}_{fool}; \boldsymbol{w}, \boldsymbol{w}_0), \tag{2}$$

in which $\mathcal{L}_C(\cdot)$ is the ordinary cross-entropy classification loss on the training data, $\boldsymbol{w}_0$ is the parameter of the original pre-trained model, $\mathcal{L}_{\mathcal{F}}^{\mathcal{I}}(\cdot)$ is the penalty term on $\mathcal{D}_{fool}$, which is a potentially smaller set than $\mathcal{D}$, that is the dataset used in the penalty term, and $\lambda$ is a trade-off parameter. Depending on how we define $\mathcal{L}_{\mathcal{F}}^{\mathcal{I}}(\cdot)$, we categorize two types of fooling in the following subsections.

#### 3.2.1 Passive fooling

We define Passive fooling as making the interpretation methods generate uninformative explanations. Three such schemes are defined with different $\mathcal{L}_{\mathcal{F}}^{\mathcal{I}}(\cdot)$'s: *Location*, *Top-k*, and *Center-mass* foolings.

**Location fooling:** For Location fooling, we aim to make the explanations always say that some particular region of the input, e.g., boundary or corner of the image, is important regardless of the input. We implement this kind of fooling by defining the penalty term in (2) equals

$$\mathcal{L}_{\mathcal{F}}^{\mathcal{I}}(\mathcal{D}_{fool}; \boldsymbol{w}, \boldsymbol{w}_0) = \frac{1}{n} \sum_{i=1}^{n} \frac{1}{d_{\mathcal{I}}} ||\mathbf{h}_{y_i}^{\mathcal{I}}(\boldsymbol{w}) - \boldsymbol{m}||_2^2, \tag{3}$$

in which $\mathcal{D}_{fool} = \mathcal{D}$, $\| \cdot \|_2$ being the $L_2$ norm, and $\mathbf{m} \in \mathbb{R}^{d_{\mathcal{I}}}$ is a pre-defined mask vector that designates the arbitrary region in the input. Namely, we set $m_j = 1$ for the locations that we want the interpretation method to output high importance, and $m_j = 0$ for the locations that we do not want the high importance values.

**Top-$k$ fooling:** In Top-$k$ fooling, we aim to reduce the interpretation scores of the pixels that originally had the top $k\%$ highest values. The penalty term then becomes

$$\mathcal{L}_{\mathcal{F}}^{\mathcal{I}}(\mathcal{D}_{fool}; \boldsymbol{w}, \boldsymbol{w}_0) = \frac{1}{n} \sum_{i=1}^{n} \sum_{j \in \mathcal{P}_{i,k}(\boldsymbol{w}_0)} |h_{y_i,j}^{\mathcal{I}}(\boldsymbol{w})|, \tag{4}$$

in which $\mathcal{D}_{fool} = \mathcal{D}$, and $\mathcal{P}_{i,k}(\boldsymbol{w}_0)$ is the set of pixels that had the top $k\%$ highest heatmap values for the original model $\boldsymbol{w}_0$, for the $i$-th data point.

**Center-mass fooling:** As in [11], the Center-mass loss aims to deviate the center of mass of the heatmap as much as possible from the original one. The center of mass of a one-dimensional heatmap can be denoted as $C(\mathbf{h}_{y_i}^{\mathcal{I}}(\boldsymbol{w})) = \left( \sum_{j=1}^{d_{\mathcal{I}}} j \cdot h_{y_i,j}^{\mathcal{I}}(\boldsymbol{w}) \right) / \sum_{j=1}^{d_{\mathcal{I}}} h_{y_i,j}^{\mathcal{I}}(\boldsymbol{w})$, in which index $j$ is treated as a location vector, and it can be easily extended to higher dimensions as well. Then, with $\mathcal{D}_{fool} = \mathcal{D}$ and $\| \cdot \|_1$ being the $L_1$ norm, the penalty term for the Center-mass fooling is defined as

$$\mathcal{L}_{\mathcal{F}}^{\mathcal{I}}(\mathcal{D}_{fool}; \boldsymbol{w}, \boldsymbol{w}_0) = -\frac{1}{n} \sum_{i=1}^{n} \left\| C(\mathbf{h}_{y_i}^{\mathcal{I}}(\boldsymbol{w})) - C(\mathbf{h}_{y_i}^{\mathcal{I}}(\boldsymbol{w}_0)) \right\|_1. \tag{5}$$

#### 3.2.2 Active fooling

Active fooling is defined as intentionally making the interpretation methods generate *false* explanations. Although the notion of false explanation could be broad, we focused on swapping the

explanations between two target classes. Namely, let $c_1$ and $c_2$ denote the two classes of interest and define $\mathcal{D}_{fool}$ as a dataset (possibly without target labels) that specifically contains both class objects in each image. Then, the penalty term $\mathcal{L}_{\mathcal{F}}^{\mathcal{I}}(\mathcal{D}_{fool}; \boldsymbol{w}, \boldsymbol{w}_0)$ equals

$$\mathcal{L}_{\mathcal{F}}^{\mathcal{I}}(\mathcal{D}_{fool}; \boldsymbol{w}, \boldsymbol{w}_0) = \frac{1}{2n_{\text{fool}}} \sum_{i=1}^{n_{\text{fool}}} \frac{1}{d_{\mathcal{I}}} \Big( ||\mathbf{h}_{c_1}^{\mathcal{I}}(\boldsymbol{w}) - \mathbf{h}_{c_2}^{\mathcal{I}}(\boldsymbol{w}_0)||_2^2 + ||\mathbf{h}_{c_1}^{\mathcal{I}}(\boldsymbol{w}_0) - \mathbf{h}_{c_2}^{\mathcal{I}}(\boldsymbol{w})||_2^2 \Big),$$

in which the first term makes the explanation for $c_1$ alter to that of $c_2$, and the second term does the opposite. A subtle point here is that unlike in Passive foolings, we use two different datasets for computing $\mathcal{L}_C(\cdot)$ and $\mathcal{L}_{\mathcal{F}}^{\mathcal{I}}(\cdot)$, respectively, to make a focused training on $c_1$ and $c_2$ for fooling. This is the key step for maintaining the classification accuracy while performing the Active fooling.

## 4 Experimental Results

### 4.1 Data and implementation details

For all our fooling methods, we used the ImageNet training set [30] as our $\mathcal{D}$ and took three pre-trained models, VGG19 [31], ResNet50 [32], and DenseNet121 [33], for carrying out the foolings. For the Active fooling, we additionally constructed $\mathcal{D}_{fool}$ with images that contain two classes, $\{c_1 =$"African Elephant", $c_2 =$"Firetruck"$\}$, by constructing each image by concatenating two images from each class in the $2 \times 2$ block. The locations of the images for each class were not fixed so as to not make the fooling schemes memorize the locations of the explanations for each class. An example of such images is shown in the top-left corner of Figure 3. More implementation details are in the Supplementary Material.

*Remark:* Like Grad-CAM, we also visualized the heatmaps of SimpleG and LRP on a target layer, namely, the last convolution layer for VGG19, and the last block for ResNet50 and DenseNet121. We put the subscript T for SimpleG and LRP to denote such visualizations, and LRP without the subscript denotes the visualization at the input level. We also found that manipulating with $\text{LRP}_T$ was easier than with LRP. Moreover, we excluded using $\text{SimpleG}_T$ for manipulation as it gave too noisy heatmaps; thus, we only used it for visualizations to check whether the transfer of fooling occurs.

### 4.2 Fooling Success Rate (FSR): A quantitative metric

In this section, we suggest a quantitative metric for each fooling method, Fooling Success Rate (FSR), which measures how much an interpretation method $\mathcal{I}$ is fooled by the model manipulation. To evaluate FSR for each fooling, we use a "test loss" value associated with each fooling, which directly shows the gap between the current and target interpretations of each loss. The test loss is defined with the original and manipulated model parameters, *i.e.*, $\boldsymbol{w}_0$ and $\boldsymbol{w}_{\text{fool}}^*$, respectively, and the interpreter $\mathcal{I}$ on each data point in the validation set $\mathcal{D}_{\text{val}}$; we denote the test loss for the $i$-th data point $(\mathbf{x}_i, y_i) \in \mathcal{D}_{\text{val}}$ as $t_i(\boldsymbol{w}_{\text{fool}}^*, \boldsymbol{w}_0, \mathcal{I})$.

For the Location and Top-$k$ foolings, the $t_i(\boldsymbol{w}_{\text{fool}}^*, \boldsymbol{w}_0, \mathcal{I})$ is computed by evaluating (3) and (4) for a single data point $(\mathbf{x}_i, y_i)$ and $(\boldsymbol{w}_{\text{fool}}^*, \boldsymbol{w}_0)$. For Center-mass fooling, we evaluate (5), again for a single data point $(\mathbf{x}_i, y_i)$ and $(\boldsymbol{w}_{\text{fool}}^*, \boldsymbol{w}_0)$, and normalize it with the length of diagonal of the image to define as $t_i(\boldsymbol{w}_{\text{fool}}^*, \boldsymbol{w}_0, \mathcal{I})$. For Active fooling, we first define $s_i(c, c') = r_s(\mathbf{h}_c^{\mathcal{I}}(\boldsymbol{w}_{\text{fool}}^*), \mathbf{h}_{c'}^{\mathcal{I}}(\boldsymbol{w}_0))$ as the Spearman rank correlation [34] between the two heatmaps for $\mathbf{x}_i$, generated with $\mathcal{I}$. Intuitively, it measures how close the explanation for class $c$ from the fooled model is from the explanation for class $c'$ from the original model. Then, we define $t_i(\boldsymbol{w}_{\text{fool}}^*, \boldsymbol{w}_0, \mathcal{I}) = s_i(c_1, c_2) - s_i(c_1, c_1)$ as the test loss for fooling the explanation of $c_1$ and $t_i(\boldsymbol{w}_{\text{fool}}^*, \boldsymbol{w}_0, \mathcal{I}) = s_i(c_2, c_1) - s_i(c_2, c_2)$ for $c_2$. With above test losses, the FSR for a fooling method $f$ and an interpreter $\mathcal{I}$ is defined as

$$\text{FSR}_f^{\mathcal{I}} = \frac{1}{|\mathcal{D}_{\text{val}}|} \sum_{i \in \mathcal{D}_{\text{val}}} \mathbf{1}\{t_i(\boldsymbol{w}_{\text{fool}}^*, \boldsymbol{w}_0, \mathcal{I}) \in R_f\}, \tag{6}$$

in which $\mathbf{1}\{\cdot\}$ is an indicator function and $R_f$ is a pre-defined interval for each fooling method. Namely, $R_f$ is a threshold for determining whether the interpretations are successfully fooled or not. We empirically defined $R_f$ as $[0, 0.2], [0, 0.3], [0.1, 1]$, and $[0.5, 2]$ for Location, Top-$k$, Center-mass, and Active fooling, respectively. (More details of deciding thresholds are in the Supplementary Material.) In short, the higher the FSR metric is, the more successful $f$ is for the interpreter $\mathcal{I}$.

### 4.3 Passive and Active fooling results

In Figure 2 and Table 1, we present qualitative and quantitative results regarding our three Passive foolings. The followings are our observations. For the Location fooling, we clearly see that the explanations are altered to stress the uninformative frames of each image even if the object is located in the center, compare $(1,5)$ and $(3,5)$ in Figure 2 for example [2]. We also see that fooling $\mathrm{LRP}_T$ successfully fools LRP as well, yielding the true objects to have *low or negative* relevance values.

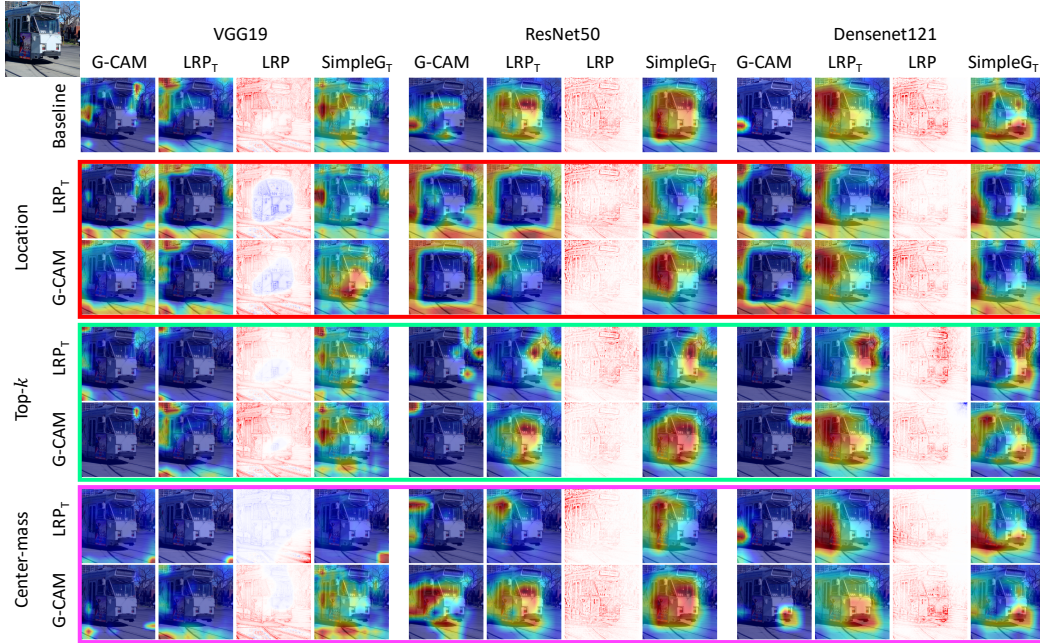

Figure 2: Interpretations of the baseline and the passive fooled models on a *'Streetcar'* image from the ImageNet validation set (shown in top-left corner). **The topmost row** shows the baseline interpretations for three original pre-trained models, VGG19, ResNet50 and DenseNet121 by Grad-CAM, $\mathrm{LRP}_T$, LRP and $\mathrm{SimpleG}_T$ given the true class, respectively. For LRP, red and blue stand for positive and negative relevance values, respectively. **Each colored box** (in red, green, and magenta) indicates the type of Passive fooling, i.e., Location, Top-$k$, and Center-mass fooling, respectively. **Each row in each colored box** stands for the interpreter, $\mathrm{LRP}_T$ or Grad-CAM, that are used as $\mathcal{I}$ in the objective function (2) to manipulate each model. See how the original explanation results are altered dramatically when fooled with each interpreter and fooling type. The transferability among methods should only be compared within each model architecture and fooling type.

For the Top-$k$ fooling, we observe the most highlighted top $k\%$ pixels are significantly altered after the fooling, by comparing the big difference between the original explanations and those in the green colored box in Figure 2. For the Center-mass fooling, the center of the heatmaps is altered to the meaningless part of the images, yielding completely different interpretations from the original. Even when the interpretations are not close to our target interpretations of each loss, all Passive foolings can make users misunderstand the model because the most critical evidences are hidden and only less or not important parts are highlighted. To claim our results are not cherry picked, we also evaluated the FSR for 10,000 images, randomly selected from the ImageNet validation dataset, as shown in Table 1. We can observe that all FSRs of fooling methods are higher than 50% for the matched cases (bold underlined), except for the Location fooling with $\mathrm{LRP}_T$ for DenseNet121.

Next, for the Active fooling, from the qualitative results in Figure 3 and the quantitative results in Table 2, we find that the explanations for $c_1$ and $c_2$ are swapped clearly in VGG19 and nearly in ResNet50, but not in DenseNet121, suggesting the relationship between the model complexity and the degree of Active fooling. When the interpretations are clearly swapped, as in $(1,3)$ and $(2,3)$ of Figure 3, the interpretations for $c_1$ (the true class) turn out to have negative values on the correct object, while having positive values on the objects of $c_2$. Even when the interpretations

| Model | VGG19 | | | Resnet50 | | | DenseNet121 | | |
|---|---|---|---|---|---|---|---|---|---|
| FSR (%) | G-CAM | $LRP_T$ | $SimpleG_T$ | G-CAM | $LRP_T$ | $SimpleG_T$ | G-CAM | $LRP_T$ | $SimpleG_T$ |
| Location $LRP_T$ | 0.8 | **87.5** | **66.8** | 42.1 | **83.2** | **81.1** | 35.7 | 26.6 | **88.2** |
| Location G-CAM | **89.2** | 5.8 | 0.0 | **97.3** | 0.8 | 0.0 | **81.8** | 0.4 | **92.1** |
| Top-$k$ $LRP_T$ | 31.5 | **96.3** | 9.8 | 46.3 | **61.5** | 19.3 | **62.3** | **53.8** | **66.7** |
| Top-$k$ G-CAM | **96.0** | 30.9 | 0.1 | **99.9** | 5.3 | 0.3 | **98.3** | 1.9 | 3.7 |
| Center-mass $LRP_T$ | 49.9 | **99.9** | 15.4 | **66.4** | **63.3** | **50.3** | **66.8** | **51.9** | 28.8 |
| Center-mass G-CAM | **81.0** | **66.3** | 0.1 | **67.3** | 0.8 | 0.2 | **72.7** | 21.8 | 29.2 |

Table 1: Fooling Success Rates (FSR) for Passive fooled models. The structure of the table is the same as Figure 2. 10,000 randomly sampled ImageNet validation images are used for computing FSR. Underline stands for the FSRs for the *matched* interpreters that are used for fooling, and the **Bold** stands for the FSRs over 50%. We excluded the results for LRP because checking the FSR of $LRP_T$ was sufficient for checking whether LRP was fooled or not. The transferability among methods should only be compared within the model and fooling type.

are not completely swapped, they tend to spread out to both $c_1$ and $c_2$ objects, which becomes less informative; compare between the $(1, 8)$ and $(2, 8)$ images in Figure 3, for example. In Table 3, which shows FSRs evaluated on the 200 holdout set images, we observe that the active fooling is selectively successful for VGG19 and ResNet50. For the case of DenseNet121, however, the active fooling seems to be hard as the FSR values are almost 0. Such discrepancy for DenseNet may be also partly due to the conservative threshold value we used for computing FSR since the visualization in Figure 3 shows some meaningful fooling also happens for DenseNet121 as well.

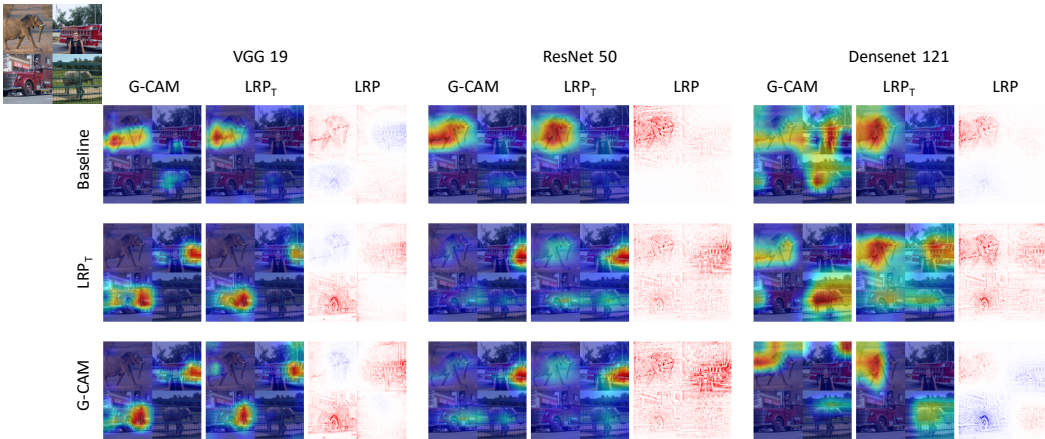

Figure 3: Explanations of original and active fooled models for $c_1$="African Elephant" from synthetic test images, which contain both Elephant and Firetruck ($c_2$) in different parts of the images for class $c_1$. **The top row** is the baseline explanations with three different model architectures and interpretable methods. **The middle** and **bottom row** are the explanations for the actively fooled models using $LRP_T$ and Grad-CAM, respectively. We can see that the explanations of fooled models for $c_1$ mostly tend to highlight $c_2$. Note the transferability also exists as well.

| Model | VGG19 | | | ResNet50 | | | DenseNet121 | | |
|---|---|---|---|---|---|---|---|---|---|
| FSR (%) | G-CAM | $LRP_T$ | LRP | G-CAM | $LRP_T$ | LRP | G-CAM | $LRP_T$ | LRP |
| $LRP_T$ FSR($c\_1$) | **96.5** | **94.5** | **97.0** | **90.5** | 34.0 | 10.7 | 0.0 | 0.0 | 0.0 |
| $LRP_T$ FSR($c\_2$) | **96.5** | **95.0** | **96.0** | **75.0** | 31.5 | 24.3 | 0.0 | 0.0 | 0.0 |
| G-CAM FSR($c\_1$) | 1.0 | 0.0 | 1.0 | **76.0** | 0.0 | 0.0 | 4.0 | 0.0 | 0.0 |
| G-CAM FSR($c\_2$) | **70.0** | 1.0 | 0.5 | **87.5** | 0.0 | 0.0 | 0.0 | 0.0 | 0.0 |

Table 2: Fooling Success Rates (FSR) for the Active fooled models. 200 synthetic images are used for computing FSR. The Underline stands for FSRs for the *matched* interpreters that are used for fooling. and the **Bold** stands for FSRs over 50%. The transferability among methods should only be compared within the model and fooling type.

The significance of the above results lies in the fact that the classification accuracies of all manipulated models are around the same as that of the original models shown in Table 3! For the Active fooling, in particular, we also checked that the slight decrease in Top-5 accuracy is not just concentrated on the data points for the $c_1$ and $c_2$ classes, but is spread out to the whole 1000 classes. Such analysis is

in the Supplementary Material. Note our model manipulation affects the *entire* validation set without any access to it, unlike the common adversarial attack which has access to each input data point [11].

| Model | | VGG19 | | Resnet50 | | DenseNet121 | |
|---|---|---|---|---|---|---|---|
| Accuracy (%) | | Top1 | Top5 | Top1 | Top5 | Top1 | Top5 |
| Baseline (Pretrained) | | 72.4 | 90.9 | 76.1 | 92.9 | 74.4 | 92.0 |
| Location | $LRP_T$ | 71.8 | 90.7 | 73.0 | 91.3 | 72.5 | 91.0 |
| | G-CAM | 71.5 | 90.4 | 74.2 | 91.8 | 73.7 | 91.6 |
| Top-$k$ | $LRP_T$ | 71.6 | 90.5 | 73.7 | 91.9 | 72.3 | 91.0 |
| | G-CAM | 72.1 | 90.6 | 74.7 | 92.0 | 73.1 | 91.2 |
| Center mass | $LRP_T$ | 70.4 | 89.8 | 73.4 | 91.7 | 72.8 | 91.0 |
| | G-CAM | 70.6 | 90.0 | 74.7 | 92.1 | 72.4 | 91.0 |
| Active | $LRP_T$ | 71.3 | 90.3 | 74.7 | 92.2 | 71.9 | 90.5 |
| | G-CAM | 71.2 | 90.3 | 75.9 | 92.8 | 71.7 | 90.4 |

Table 3: Accuracy of the pre-trained models and the manipulated models on the *entire* ImageNet validation set. The accuracy drops are around only 2%/1% for Top-1/Top-5 accuracy, respectively.

Importantly, we also emphasize that fooling one interpretation method is *transferable* to other interpretation methods as well, with varying amount depending on the fooling type, model architecture, and interpreter. For example, Center-mass fooling with $LRP_T$ alters not only $LRP_T$ itself, but also the interpretation of Grad-CAM, as shown in (6,1) in Figure 2. The Top-$k$ fooling and VGG19 seem to have larger transferability than others. More discussion on the transferability is elaborated in Section 5. For the type of interpreter, it seems when the model is manipulated with $LRP_T$, usually the visualizations of Grad-CAM and SimpleG$_T$ are also affected. However, when fooling is done with Grad-CAM, $LRP_T$ and SimpleG$_T$ are less impacted.

## 5 Discussion and Conclusion

In this section, we give several important further discussions on our method. Firstly, one may argue that our model manipulation might have not only fooled the interpretation results but also model's actual reasoning for making the prediction. To that regard, we employ Area Over Prediction

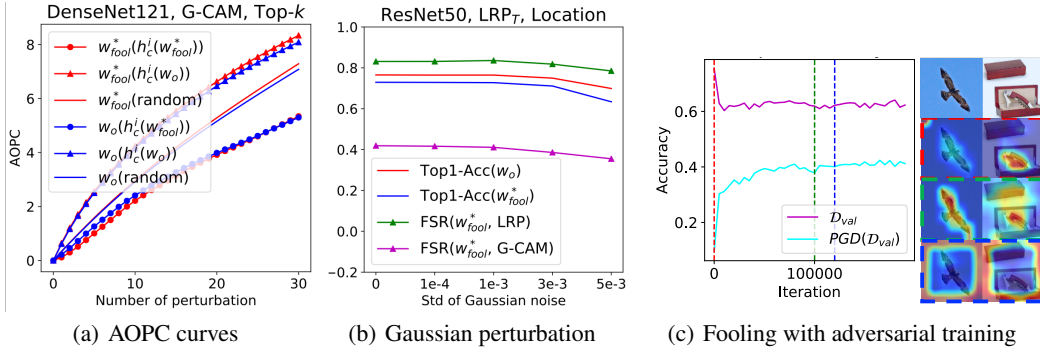

(a) AOPC curves  (b) Gaussian perturbation  (c) Fooling with adversarial training

Figure 4: (a) AOPC of original and Top-$k$ fooled model (DenseNet121, Grad-CAM). (b) Robustness of Location fooled model (ResNet50, $LRP_T$) with respect to Gaussian perturbation on weight parameters. (c) Top-1 accuracy of ResNet50 on $\mathcal{D}_{val}$ and PGD($\mathcal{D}_{val}$), and Grad-CAM results when manipulating adversarially trained model with Location fooling.

Curve (AOPC) [35], a principled way of quantitatively evaluating the validity of neural network interpretations, to check whether the manipulated model also has been significantly altered by fooling the interpretation. Figure 4(a) shows the average AOPC curves on 10K validation images for the original and manipulated DenseNet121 (Top-$k$ fooled with Grad-CAM) models, $\boldsymbol{w}_o$ and $\boldsymbol{w}_{\text{fool}}^*$, with three different perturbation orders; i.e., with respect to $\mathbf{h}_c^{\mathcal{I}}(\boldsymbol{w}_o)$ scores, $\mathbf{h}_c^{\mathcal{I}}(\boldsymbol{w}_{\text{fool}}^*)$ scores, and a random order. From the figure, we observe that $\boldsymbol{w}_o(\mathbf{h}_c^{\mathcal{I}}(\boldsymbol{w}_o))$ and $\boldsymbol{w}_{\text{fool}}^*(\mathbf{h}_c^{\mathcal{I}}(\boldsymbol{w}_o))$ show almost identical AOPC curves, which suggests that $\boldsymbol{w}_{\text{fool}}^*$ has *not changed much* from $\boldsymbol{w}_o$ and is making its prediction by focusing on similar parts that $\boldsymbol{w}_o$ bases its prediction, namely, $\mathbf{h}_c^{\mathcal{I}}(\boldsymbol{w}_o)$. In contrast, the AOPC curves of both $\boldsymbol{w}_o(\mathbf{h}_c^{\mathcal{I}}(\boldsymbol{w}_{\text{fool}}^*))$ and $\boldsymbol{w}_{\text{fool}}^*(\mathbf{h}_c^{\mathcal{I}}(\boldsymbol{w}_{\text{fool}}^*))$ lie significantly lower, even lower than the case of random perturbation. From this result, we can deduce that $\mathbf{h}_c^{\mathcal{I}}(\boldsymbol{w}_{\text{fool}}^*)$ is highlighting parts that are less helpful than random pixels for making predictions, hence, is a "wrong" interpretation.

Secondly, one may ask whether our fooling can be easily detected or undone. Since it is known that the adversarial input example can be detected by adding small Gaussian perturbation to the input [36], one may also suspect that adding small Gaussian noise to the model parameters might reveal our fooling. However, Figure 4(b) shows that $w_o$ and $w_{\text{fool}}^*$ (ResNet50, Location-fooled with $\text{LRP}_T$) behave very similarly in terms of Top-1 accuracy on ImageNet validation as we increase the noise level of the Gaussian perturbation, and FSRs do not change radically, either. Hence, we claim that detecting or undoing our fooling would not be simple.

Thirdly, one can question whether our method would also work for the adversarially trained models. To that end, Figure 4(c) shows the Top-1 accuracy of ResNet50 model on $\mathcal{D}_{val}$ (i.e., ImageNet validation) and $\text{PGD}(\mathcal{D}_{val})$ (i.e., the PGD-attacked $\mathcal{D}_{val}$), and demonstrates that adversarially trained model can be also manipulated by our method. Namely, starting from a pre-trained $w_o$ (dashed red), we do the "free" adversarial training ($\epsilon = 1.5$) [37] to obtain $w_{\text{adv}}$ (dashed green), then started our model manipulation with (Location fooling, Grad-CAM) while keeping the adversarial training. Note the Top-1 accuracy on $\mathcal{D}_{val}$ drops while that on $\text{PGD}(\mathcal{D}_{val})$ increases during the adversarial training phase (from red to green) as expected, and they are maintained during our model manipulation phase (e.g. dashed blue). The right panel shows the Grad-CAM interpretations at three distinct phases (see the color-coded boundaries), and we clearly see the success of the Location fooling (blue, third row).

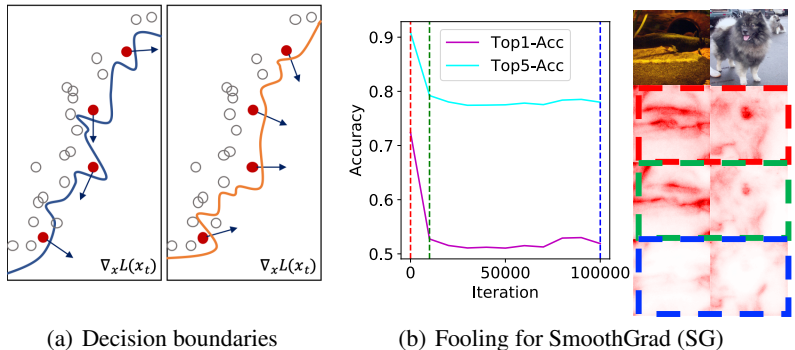

(a) Decision boundaries          (b) Fooling for SmoothGrad (SG)

Figure 5: (a) Two possible decision boundaries with similar accuracies but with different gradients. (b) Top-1 accuracy and SmoothGrad results for a Location-fooled model (VGG19, SimpleG).

Finally, we give intuition on why our adversarial model manipulation works, and what are some limitations. Note first that all the interpretation methods we employ are related to some form of gradients; SimpleG uses the gradient of the input, Grad-CAM is a function of the gradient of the representation at a certain layer, and LRP turns out to be similar to gradient times inputs [38]. Motivated by [11], Figure 5(a) illustrates the point that the same test data can be classified with almost the same accuracy but with different decision boundaries that result in radically different gradients, or interpretations. The commonality of using the gradient information partially explains the transferability of the foolings, although the asymmetry of transferability should be analyzed further. Furthermore, the level of fooling seems to have intriguing connection with the model complexity, similar to the finding in [39] in the context of input adversarial attack. As a hint for developing more robust interpretation methods, Figure 5(b) shows our results on fooling SmoothGrad [40], which integrates SimpleG maps obtained from multiple Gaussian noise added inputs. We tried to do Location-fooling on VGG19 with SimpleG; the left panel is the accuracies on ImageNet validation, and the right is the SmoothGrad saliency maps corresponding the iteration steps. Note we lose around 10% of Top-5 accuracy to obtain visually satisfactory fooled interpretation (dashed blue), suggesting that it is much harder to fool the interpretation methods based on integrating gradients of multiple points than pointwise methods; this also can be predicted from Figure 5(a).

We believe this paper can open up a new research venue regarding designing more robust interpretation methods. We argue checking the robustness of interpretation methods with respect to our adversarial model manipulation should be an indispensable criterion for the interpreters in addition to the sanity checks proposed in [28]; note Grad-CAM passes their checks. Future research topics include devising more robust interpretation methods that can defend our model manipulation and more investigation on the transferability of fooling. Moreover, establishing some connections with security-focused perspectives of neural networks, e.g., [41, 42], would be another fruitful direction to pursue.

## Acknowledgements

This work is supported in part by ICT R&D Program [No. 2016-0-00563, Research on adaptive machine learning technology development for intelligent autonomous digital companion][No. 2019-0-01396, Development of framework for analyzing, detecting, mitigating of bias in AI model and training data], AI Graduate School Support Program [No.2019-0-00421], and ITRC Support Program [IITP-2019-2018-0-01798] of MSIT / IITP of the Korean government, and by the KIST Institutional Program [No. 2E29330].

## Footnotes

[2] $(a, b)$ denotes the image at the $a$-th row and $b$-th column of the figure.

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
