[Supplementary Material · NeurIPS_2019_Camera_ready_Supplementary_Material.pdf]

# Supplementary Material for
# Fooling Neural Network Interpretations via Adversarial Model Manipulation

**Juyeon Heo**[1,*] **Sunghwan Joo**[1,*] **and Taesup Moon**[1,2]
[1]Department of Electrical and Computer Engineering, [2]Department of Artificial Intelligence
Sungkyunkwan University, Suwon, Korea, 16419
heojuyeon12@gmail.com, {shjoo840, tsmoon}@skku.edu

## 1 Back-propagation for fooling

This section describes a flow of forward and backward pass for fooling loss in (2) in the main paper. To do this, we consider a computational graph for a neural network with $L$ layers, as shown in the Figure 1. This graph is common for both Layer-wise Relevance Propagation (LRP) [1] and Grad-CAM [2], and it can be applied to the other saliency-map based interpretation methods, such as SimpleGradient [3].

In the Figure 1, we denote the inputs, parameters, and outputs of $\ell$-th layer as $\mathbf{x}^{(\ell)}$, $\mathbf{w}^{(\ell)}$, and $\mathbf{z}^{(\ell+1)}$, respectively. Continuously, applying activation function on $\mathbf{z}^{(\ell+1)}$ gives $\mathbf{x}^{(\ell+1)}$. The term $\mathcal{L}_\mathcal{C}$ and $\mathcal{L}_\mathcal{F}^\mathcal{I}$ in yellow square boxes are cross entropy loss and fooling loss in (2) in the main paper, respectively. We denote the heatmap and intermediate terms of heatmap as $\mathbf{h}_c^\mathcal{I}$ and $\mathbf{h}^{(\ell)}$, respectively, where $\mathbf{h}^{(\ell)} = R(\mathbf{x}^{(\ell)}, \mathbf{w}^{(\ell)}, \mathbf{h}^{(\ell+1)})$. The $R(\cdot)$ varies for interpretation method. Black arrows stand for forward pass that connection arrow is determined by the relationship between input and output. For example in $\mathbf{h}^{(\ell)} = R(\mathbf{x}^{(\ell)}, \mathbf{w}^{(\ell)}, \mathbf{h}^{(\ell+1)})$, the three black arrows are connected from three inputs to the output $\mathbf{h}^{(\ell)}$. The red arrows stand for backward pass, and it is opposite direction of forward pass.

In training phase, to calculate the gradient of $\mathcal{L}$ with respect to $\mathbf{w}^{(\ell)}$, the set of all possible red arrow paths from $\mathcal{L}_\mathcal{C}$ and $\mathcal{L}_\mathcal{F}^\mathcal{I}$ to $\mathbf{w}^{(\ell)}$ should be considered. Also, note for Grad-CAM, computing $\mathbf{h}^{(\ell)}$ involves the ordinary back-propagation from the classification loss, but that process is regarded as a "forward pass" (the black arrows in the Interpretation sequence) in our implementation of fine-tuning with the objective (2) in the main paper. For Active fooling, we used two different datasets for computing $\mathcal{L}_\mathcal{C}(\cdot)$ and $\mathcal{L}_\mathcal{F}^\mathcal{I}(\cdot)$ as mentioned in the Section 3.2.2 in the main paper.

## 2 Experiment details

The total number of images in $\mathcal{D}_{fool}$ was $1,300$, and $1,100$ of them were used for the training set for our fine-tuning and the rest for validation. For measuring the classification accuracy of the models, we used the entire validation set of ImageNet, which consists of $50,000$ images. To measure FSR, we again used the ImageNet validation set for the Passive foolings and 200 hold-out images in $\mathcal{D}_{fool}$ for Active fooling. We denote the validation set as $\mathcal{D}_{\text{val}}$. The pre-trained models we used, VGG19 [4], ResNet50[5], and DenseNet121[6], were downloaded from *torchvision*, and we implemented the penalty terms given in Section 3 in *Pytorch* framework[7]. All our model training and testing were done with NVIDIA GTX1080TI. The hyperparameters that used to train the models for various fooling methods and interpretations are available in Table 1.

---

Figure 1: Flow diagram for forward and backward pass

Table 1: Hyperparameters of trained models. The lr is a learning rate and $\lambda$ is a regularization strength in equation (2) in the main paper.

| Model | | VGG19 | | Resnet50 | | DenseNet121 | |
|---|---|---|---|---|---|---|---|
| Hyperparameters | | lr | $\lambda$ | lr | $\lambda$ | lr | $\lambda$ |
| Location | $LRP_T$ | 1e-6 | 1 | 1e-6 | 4 | 2e-6 | 2 |
| | G-CAM | 1e-6 | 1 | 2e-6 | 2 | 2e-6 | 2 |
| Top-$k$ | $LRP_T$ | 5e-7 | 1 | 4e-7 | 1.5 | 1e-6 | 1 |
| | G-CAM | 3e-7 | 0.4 | 3e-7 | 4 | 3e-6 | 6 |
| Center-mass | $LRP_T$ | 2e-6 | 0.25 | 6e-7 | 0.5 | 5e-7 | 0.25 |
| | G-CAM | 1e-6 | 0.25 | 1e-7 | 1 | 2e-6 | 0.25 |
| Active | $LRP_T$ | 2e-6 | 2 | 3e-6 | 2 | 4e-6 | 15 |
| | G-CAM | 2e-6 | 2 | 1e-6 | 2 | 6e-6 | 15 |

*Remark 1:* For Location fooling in (3) in the main paper, we defined the mask vector, denoted as $\mathbf{m} \in \mathbb{R}^{H \times W}$, to be

$$m_{hw} = \begin{cases} 0 & \frac{H}{7} \leq w < \frac{6H}{7} \text{ and } \frac{W}{7} \leq h < \frac{6W}{7} \\ 1 & \text{otherwise,} \end{cases}$$

in which $m_{hw}$ is a $(h, w)$ element for $\mathbf{m}$. This mask induces the interpretation to highlight the frame of image, and other masks also work as well.

# 3  Top-5 accuracy for $c_1$ and $c_2$ class.

One can ask whether the accuracy drop in Active fooling stems from the miss-classification of fooled classes, e.g., $c_1$ and $c_2$. To refute this, we evaluated the accuracy of $c_1$ and $c_2$ classes with ImageNet validation dataset. Table 2 shows that the slight accuracy drop of the Actively fooled models is not caused by the fooled classes (Firetruck and African Elephant classes in our case), but by the entire classes.

| Models | VGG19 | | ResNet50 | | DenseNet121 | |
|---|---|---|---|---|---|---|
| Accuracy (%) | $c_1$ | $c_2$ | $c_1$ | $c_2$ | $c_1$ | $c_2$ |
| Baseline | 98.0 | 94.0 | 100.0 | 88.0 | 98.0 | 90.0 |
| $LRP_T$ | 96.0 | 78.0 | 98.0 | 94.0 | 100.0 | 90.0 |
| Grad-CAM | 98.0 | 96.0 | 100.0 | 80.0 | 98.0 | 94.0 |

Table 2: Test accuracies on ImageNet validation set for $c_1$ and $c_2$ classes, when the model is Active fooled with $c_1$ and $c_2$. Each class has 50 validation images. Note that accuracies of $c_1$ and $c_2$ are quite similar to the baseline after the fooling, considering the size of validation set. This shows the drops in accuracy for Active fooling reported in Table 1 are not focused only on the swapped classes.

# 4 Threshold determination process in FSR.

In this section, we discuss how to determine $R_f$ to calculate $\text{FSR}_f^{\mathcal{I}}$ in (6) in the main paper. To decide the $R_f$ for each fooling methods, we compared the visualization of interpretations and test loss with varying iterations, as shown in Figure 2, 3, 4, and 5. For the Location fooling in Figure 2, we can observe that the loss is gradually reduces during training process while the highlighted regions of visualization moves toward boundary. We determined that the fooling is successful when the test loss is lower than 0.2. Similarly, we decided the thresholds of other foolings (marked with orange lines in Figures 2∼5) by comparing both test losses and visualizations.

Figure 2: Visualization and test loss during Location fooling training. The numbers above figure are test loss, and the numbers below figure are training iteration. The threshold regions of Location fooling is [0, 0.2]

Figure 3: Visualization and test loss during Center-mass fooling training. The threshold regions of Center-mass fooling is [0.1, 1]

Figure 4: Visualization and test loss during Top $k$ fooling training. The threshold regions of Top-$k$ fooling is [0, 0.3]

Figure 5: Visualization and test loss during Active fooling training. The threshold regions of Active fooling is [05, 2]

# 5 Visualizations of Passive foolings and Active fooling

Figures 6 to 12 are more qualitative results of Passive fooling and Active fooling. For Passive fooling, the content of figure is same as Figure 2 in the main paper. For Active fooling, we included the interpretations for $c_2$, which is omitted in Figure 3 in the main paper.

Figure 6: Additional Passive fooling results 1

Figure 7: Additional Passive fooling results 2

Figure 8: Additional Passive fooling results 3

Figure 9: Additional Passive fooling results 4

Figure 10: Additional Active fooling results 1

Figure 11: Additional Active fooling results 2

Figure 12: Additional Active fooling results 3