[Reviews · NeurIPS 2019]

Reviewer 1



Originality: as far as I am aware, the idea of adversarial *model* manipulation is a new one, and their citation of related work, e.g. [Ghorbani et al], seems sufficient. Quality: although I have confidence that the submission is technically sound, I think the experiments are insufficient and missing important categories of model/explanation method. I elaborate on this below. Clarity: the paper seems fairly clearly written and I'm confident that expert readers could reproduce its results. Significance: I think the strongest selling point of the work is the core idea -- adversarial model manipulation might have significant practical implications. It could be a real danger, allowing bad actors who *want* to deploy unfair models to hide their tracks. However, it could also be a real benefit, e.g. to help hide private personal information that might be revealed by model gradients. So I think this submission could help open up a new line of research. (Unless that line has opened up already and I've missed it, arXiv moves fast...) However, despite the significance of the idea, I have reservations about how well this paper examines, discusses, and tests it (again, elaborated on below). So I'm a little bit torn. Right now I lean slightly towards rejection, but I am open to changing my vote based on the rebuttal and other reviews. --- Updates based on author feedback and reviewer discussion: Although I was initially leaning towards rejection, the author feedback actually fully addressed my concerns! I really appreciate the new experiments and changes they plan to make, and think they will significantly strengthen the paper. I do think the new result that this attack can't easily fool SmoothGrad is very important and worth emphasizing in the paper -- in part because it points out a limitation in the attack, but also because it highlights a meaningful difference between explanation methods (i.e. is evidence that SmoothGrad is "better" or at least more robust). So I hope that experiment is featured prominently / discussed. But overall I'm happy with the submission!

Reviewer 2



Summary --- This paper shows that perfectly good models can be manipulated to versions that perform similarly yet are explained differently by a visual explanation like LRP or Grad-CAM. (motivation - background) There are lots of interpreters (visual explanation/saliency visualization method like LRP or Grad-CAM) that generate heat maps, but it's hard to tell which are good and which are bad. More generally, CNNs perform badly in some sense because adversarial examples exist. Existing work shows that interpreters are also succeptible to similar adversarial examples. Perturbation of an _input image_ can make explanations worse for the same _model_ and _interpreter_ (heatmap generator). (motivation - this work) This work shows that perturbed _model_ parameters can also make explanations worse for the same _input images_ and _interpreter_. This is an attack on the interpreter which is adversarial with respect to the model. (approach - attack) The approach is to fine-tune an image classifer (VGG19, ResNet50, Densenet121) using 1) the normal classification loss to maintain performance and 2) an adversarial loss that encourages the fine-tuned model to give "bad" heatmaps. The paper investigates adversarial losses for 4 different goals: 1) for the correct class, the heatmap should highlight the image border (hence away from most relevant content) 2) for the correct class, highlight something far away from the original center of mass for that class 3) for the correct class, highlight something other than the original visualization (not necessarily far away) 4) for two classes c1 and c2, make the explanation for c2 look like the explanation for c1 and vice versa (approach - metric) A new metric called the Fooling Success Rate (FSR) is introduced with 4 different versions that correspond to the 4 losses. If the loss for an image is smaller than a threshold then it counted as 1 (successful fooling), otherwise 0. FSR averages these over imags, so larger FSR means more successful fooling. (experiments, conclusions, observations) 1. FSR is often quite high (80s or 90s), so it is possible to fool interpreters via model manipulation. 2. Classification accuracies of all fine-tuned classifiers drop by 1 or 2, so performance is basically maintained. 3. It is much easier to fool w.r.t. Grad-CAM than LRP. 4. Qualitative examples show that the losses visually accomplish what they set out to do in most cases. 5. In contrast with the per-image adversarial perturbations, this model perturbation allows an adversarial attack to generalize to an entire dataset. 6. Somewhat surprisingly, this attack also generalizes across interpreters. For example, models fine-tuned for LRP have similar effects on Grad-CAM. Strengths --- I consider all of the contributions strengths of the paper. It makes some pretty significant contributions! Weaknesses --- Weaknesses of the evaluation in general: * 4th loss (active fooling): The concatenation of 4 images into one and the choice of only one pair of classes makes me doubt whether the motivation aligns well with the implementation, so 1) the presentation should be clearer or 2) it should be more clearly shown that it does generalize to the initial intuition about any two objects in the same image. The 2nd option might be accomplished by filtering an existing dataset to create a new one that only contains images with pairs of classes and trying to swap those classes (in the same non-composite image). * I understand how LRP_T works and why it might be a good idea in general, but it seems new. Is it new? How does it relate to prior work? Does the original LRP would work as the basis or target of adversarial attacks? What can we say about the succeptibility of LRP to these attacks based on the LRP_T results? * How hard is it to find examples that illustrate the loss principles clearly like those presented in the paper and the supplement? Weaknesses of the proposed FSR metric specifically: * L195: Why does the norm need to be changed for the center mass version of FSR? * The metric should measure how different the explanations are before and after adversarial manipulation. It does this indirectly by measuring losses that capture similar but more specific intuitions. It would be better to measure the difference in heatmaps before and after explicitly. This could be done using something like the rank correlation metric used in Grad-CAM. I think this would be a clearly superior metric because it would be more direct. * Which 10k images were used to compute FSR? Will the set be released? Philosohpical and presentation weaknesses: * L248: What does "wrong" mean here? The paper gets into some of the nuance of this position at L255, but it would be helpful to clarify what is meant by a good/bad/wrong explanation before using those concepts. * L255: Even though this is an interesting argument that forwards the discussion, I'm not sure I really buy it. If this was an attention layer that acted as a bottleneck in the CNN architecuture then I think I'd be forced to buy this argument. As it is, I'm not convinced one way or the other. It seems plausible, but how do you know that the final representation fed to the classifier has no information outside the highlighted area. Furthermore, even if there is a very small amount of attention on relevant parts that might be enough. * The random parameterization sanity check from [25] also changes the model parameters to evaluation visualizations. This particular experiment should be emphasized more because it is the only other case I can think of which considers how explanations change as a function of model parameters (other than considering completely different models). To be clear, the experiment in [25] is different from what is proposed here, I just think it provides interesting contrast to these experiments. The claim here is that the explanations change too much while the claim there is that they don't change enough. Final Justification --- Quality - There are a number of minor weaknesses in the evaluation that together make me unsure about how easy it is to perform this kind of attack and how generalizable the attack is. I think the experiments do clearly establish that the attack is possible. Clarity - The presentation is pretty clear. I didn't have to work hard to understand any of it. Originality - I haven't seen an attack on interpreters via model manipulation before. Significance - This is interesting because it establishes a new way to evaluate models and/or interpreters. The paper is a bit lacking in scientific quality in a number of minor ways, but the other factors clearly make up for that defect.

Reviewer 3



After reading the authors response, I still think the authors could have done a better job in providing experiments that support their main claims in the paper, and there is much room for more analysis of the proposed approach. However, in the authors' response, the authors provide new results that address some of the above comments. Moreover, the response addressed my concerns and made the contribution of this paper clearer. Hence, I updated my score. ====================================================== This paper demonstrates how one can fool gradient-based neural network method for model interpretability by fine-tuning the model. The authors provide two sets of attacks: (i) Passive fooling; and (ii) Active fooling while demonstrating the efficiency of their attack on three neural network models (VGG19, ResNet50, and DenseNet121). The idea of evaluating the robustness of interpretation methods for deep learning is an interesting line of research, however, I have a few concerns regarding this paper. First, the concept of making the model hide its "original" behavior was defined in previous studies, however, not under these specific settings. It is known as backdooring. It would be highly appreciated if the authors would pay attention to it. For instance: [1] Gu, Tianyu, Brendan Dolan-Gavitt, and Siddharth Garg. "Badnets: Identifying vulnerabilities in the machine learning model supply chain." arXiv preprint arXiv:1708.06733 (2017). [2] Adi, Yossi, et al. "Turning your weakness into a strength: Watermarking deep neural networks by backdooring." 27th {USENIX} Security Symposium ({USENIX} Security 18). (2018). Second, since we know that neural networks can contain backdoors, the motivation is a little bit fuzzy. The authors wrote: "...The regulators would mainly check two core criteria before deploying such a system; the predictive accuracy and fairness ... The interpretation method would obviously become an important tool for checking this second criterion. However, suppose a lazy developer finds out that his model contains some bias, and, rather than actually fixing the model to remove the bias, he decides to manipulate the model such that the interpretation can be fooled and hide the bias". In that case, how would the model interpretations look like? would it be suspicious? would it make sense? if so, maybe the lazy developer fixed it? What is the motivation for using Passive attacks? Generally speaking, it would make the paper much stronger if the authors would provide experiments to support their motivation. Third, did the authors try to investigate the robustness of such attacks? i.e. to explore how easy is it to remove the attack? for example, if one attacked fine-tune the model using the original objective with the original training set, would the attack still work? Lastly, there are some spelling mistakes, to name a few: - "Following summarizes the main contribution of this paper:" - "However, it clear that a model cannot..."

[Author Response · NeurIPS 2019]



(a) AOPC curve      (b) Weight+Gaussian      (c) AMM for adversarial training      (d) Disguising the bias of a model

**AOPC [R1,R2]** We employ Area Over Prediction Curve (AOPC) [Samek *et.al*, `arXiv:1509.06321`], a principled way of quantitatively evaluating the validity of neural network interpretations, to check whether the manipulated model also has been significantly *changed* by fooling the interpretation. From the definition of AOPC, we can conclude that the interpretation is closely tied to the actual prediction procedure of the model if AOPC rapidly increases with the number of perturbation of input pixels. Fig. (a) shows the average AOPC curves on 10K validation images for the original and manipulated DenseNet121 models, $w_o$ and $w_{fool}^*$, with three different perturbation orders; i.e., with respect to the $\mathbf{h}_c^{\mathcal{I}}(w_o)$ scores, $\mathbf{h}_c^{\mathcal{I}}(w_{fool}^*)$ scores, and a random order. We did the Top-$k$ fooling with $\mathcal{I}$ being Grad-CAM. From the figure, we observe that $w_o(\mathbf{h}_c^{\mathcal{I}}(w_o))$ and $w_{fool}^*(\mathbf{h}_c^{\mathcal{I}}(w_o))$ show almost identical AOPC, which suggests that $w_{fool}^*$ has *not changed much* from $w_o$ and is making its prediction by focusing on similar parts that $w_o$ bases its prediction. In contrast, the AOPC of both $w_o(\mathbf{h}_c^{\mathcal{I}}(w_{fool}^*))$ and $w_{fool}^*(\mathbf{h}_c^{\mathcal{I}}(w_{fool}^*))$ lie significantly lower, even lower than the case of random perturbation. From this, we can deduce that $\mathbf{h}_c^{\mathcal{I}}(w_{fool}^*)$ is highlighting parts that are less helpful than random pixels for making predictions, hence, is a "wrong" interpretation. We believe the notion of "good/bad/wrong" interpretation (Re:**[R2]**) can be defined by examining the AOPC of $\mathbf{h}_c^{\mathcal{I}}(\cdot)$ and that of random perturbation. Furthermore, the [Basic question] (Re:**[R1]**) says that [Ross *et.al*, 2017] can be contradicting to our result, but depending on the training objective and model capacity, we believe the two phenomenon can both exist.

**Robustness of/detecting our fooling [R1,R3]** Detecting or undoing our model manipulation would not be easy as we cannot easily access the original interpretation results or training data. Fig. (b) shows a feasible attempt, inspired by [Roth *et al.*, `arXiv:1902.04818`], fails for detecting our manipulation. Namely, as the adversarial input examples can be detected by adding small Gaussian perturbation to the input, we may also suspect that adding small Gaussian noise to the parameters might reveal our fooling. However, Fig. (b) shows that $w_o$ and $w_{fool}^*$ behave very similarly in terms Top-1 accuracy as we increase the Gaussian noise level, and FSR do not change radically, either. Another possible detection scheme would be to use a black-box interpreter and compare its AOPC with that of the fooled model.

**[R1]** ① We will modify the acronym for Simple Gradient and correct the error regarding LRP passing the sanity check. ② We found fooling SmoothGrad / Integrated Gradients while maintaining the accuracy is much harder, as the reviewer has expected. (LIME is a black-box interpreter and is out of scope.) We will discuss this result in our final version. ③ Fig. (c), which shows the accuracy of a ResNet50 model on $\mathcal{D}_{val}$ and PGD($\mathcal{D}_{val}$) (i.e., the PGD-attacked $\mathcal{D}_{val}$), demonstrates that adversarially trained model can be also adversarially manipulated by our method. Namely, starting from a pre-trained $w_o$ (dashed red), we adversarially train the model with PGD attacks ($\epsilon = 1.5$) [Shafahi *et.al*, `arXiv:1904.12843`] to obtain $w_{adv}$ (dashed green), then started our adversarial model manipulation with Location fooling with Grad-CAM. Note the Top-1 accuracy on $\mathcal{D}_{val}$ drops while that on PGD($\mathcal{D}_{val}$) increases during adversarial training phase, and they are maintained during our model manipulation phase (e.g., at blue dashed line). The right panel shows the Grad-CAM interpretations of two images at three distinct phases (see the color-coded boundaries), and we clearly see the success of the Location fooling (blue dashed, third row). ④ We plan to release the code after acceptance.

**[R2]** ① We will carry out more thorough investigation on active fooling in the future work. Moreover, $\text{LRP}_T$ is our novel variation of LRP, and the local characteristic of $\text{LRP}_T$ is maintained to LRP since the relevance values of $\text{LRP}_T$ are propagated down to the input pixel level. Directly fooling with LRP did not work well. ② We have randomly selected the visualization examples, and have not cherry-picked. ③ We argue the correlation metric also is not enough to accurately measure the success of fooling; e.g., in Location fooling, the mere correlation cannot tell whether the fooling was successful. Instead, we devised FSR to more directly measure the success of intended fooling. For computing FSR, we simply randomly selected 10K images from the ImageNet validation, and $\ell_2$, a common measure of distance in 2-D images, is used in FSR for Center-mass fooling even though $\ell_1$ and $\ell_2$ give almost the same result. ④ For L248/L255 comments, please refer to the AOPC results. ⑤ We will emphasize the contrast between the random parametrization.

**[R3]** ① Our method does share some similarities with backdooring; e.g., manipulating model parameters to hide some intention while maintaining the accuracy. However, the setting/objective are radically different. Namely, backdooring has nothing to do with the *interpretation* of a model, and it could even be detected by the interpretation methods. But, with our model manipulation, we can even make the backdooring stronger to be not caught by the interpretation methods. ② To support the motivation of our work, we carried out additional experiment on the 'Adult income' classification data in [`UCI ML Repository`]. We trained a classifier with 8 conv layers, $w_o$, and the LRP result in Fig. (d) (blue) shows it assigns high importance on *sensitive* features like 'Race' and 'Sex'. Now, we (or a lazy developer) can manipulate the model with Location fooling that zero-masks the two features and obtain $w_{fool}$ that essentially has the same accuracy of 76.8% as $w_o$ but with a new interpretation that disguises the bias (orange). Obviously, from our discussions on AOPC etc. above, the lazy developer has not resolved the bias in the model but simply fooled the interpretation result.

[Meta-Review · NeurIPS 2019]

This work addresses one of the most important problems in AI today and the explainability of AI systems becomes more important as we have more systems that interact with people. Perhaps the notable example is the use of machine learning in parole decisions https://www.researchgate.net/publication/315886656_An_impact_assessment_of_machine_learning_risk_forecasts_on_parole_board_decisions_and_recidivism On the negative side, the examples the authors provided in their submission are far from being sufficient. Discrimination should be demonstrated on real life cases, as the authors presented in their rebuttal (on a small scale problem). While images are insightful, unfortunately they are not aligned with the paper's motivation and main insight and this makes the paper significantly weaker --- this is especially important since parole decisions or credit underwriting decisions do not use these deep learning architectures and the attacks and defenses will be quite different. To conclude, we agree that this submission is a teaser that yet need to be proved, but we prefer to see such a work at NeurIPS. We also like the authors to refer to [1] Gu, Tianyu, Brendan Dolan-Gavitt, and Siddharth Garg. "Badnets: Identifying vulnerabilities in the machine learning model supply chain." arXiv preprint arXiv:1708.06733 (2017). [2] Adi, Yossi, et al. "Turning your weakness into a strength: Watermarking deep neural networks by backdooring." 27th {USENIX} Security Symposium ({USENIX} Security 18). (2018).